# Developmental hematopoietic stem cell variation explains clonal hematopoiesis later in life

Jesse Kreger [1], Jazlyn A. Mooney [1], Darryl Shibata [2] & Adam L. MacLean [1]✉

Clonal hematopoiesis becomes increasingly common with age, but its cause is enigmatic because driver mutations are often absent. Serial observations infer weak selection indicating variants are acquired much earlier in life with unexplained initial growth spurts. Here we use fluctuating CpG methylation as a lineage marker to track stem cell clonal dynamics of hematopoiesis. We show, via the shared prenatal circulation of monozygotic twins, that weak selection conferred by stem cell variation created before birth can reliably yield clonal hematopoiesis later in life. Theory indicates weak selection will lead to dominance given enough time and large enough population sizes. Human hematopoiesis satisfies both these conditions. Stochastic loss of weakly selected variants is naturally prevented by the expansion of stem cell lineages during development. The dominance of stem cell clones created before birth is supported by blood fluctuating CpG methylation patterns that exhibit low correlation between unrelated individuals but are highly correlated between many elderly monozygotic twins. Therefore, clonal hematopoiesis driven by weak selection in later life appears to reflect variation created before birth.

Hematopoietic stem cells (HSCs) maintain the blood system throughout life[1]. As humans age, clonal expansions of HSCs are frequently observed[2–7]. Such clonal hematopoiesis (CH) is associated with increased risks of hematopoietic neoplasia and other diseases[8–14]. Whereas in some cases driver mutations can be found[15–17], often CH is lacking identifiable drivers that can explain their expansions.

CH is often identified by specific somatic mutations[8]. Here we take a broader view, defining CH as clonal expansions in the HSC pool that lead to a loss of stem cell diversity. Consistent with the frequent lack of strong driver mutations, serial observations of CH are often compatible with weak selection (potentially driven by epigenetic changes) because clone sizes are stable or expanding slowly over many years[2]. Such weak selection is problematic to explain CH because most somatic cells that acquire such a weak selective advantage would randomly drift out of the population and are thus unlikely to ever attain detectable frequencies[18,19].

Weak selection and slow expansion suggest that HSC subclones can arise early in life, potentially even before birth, with explained initial growth spurts[1,15,20–22]. Early acquisition of a selective advantage eventually leading to CH would explain two open questions about CH without the need for identifiable driver mutations. First, following classical evolutionary dynamics theory and its extensions[18,23–30], variants with weak selective advantages can become dominant given relatively large populations sizes, population turnover, and enough time. These conditions are present for human hematopoiesis, which exhibits a large HSC population size[31], competition between stem cells, and many decades of life. HSC variants acquired very early in life maximize the time needed for their subclones to reach dominance. Second, variants with weak selective advantages are uniquely protected from random loss and extinction during development due to the natural expansion of all HSC subclones at this time of growth. The interval before birth maximizes the possible time and HSC expansion.

[1]Department of Quantitative and Computational Biology, University of Southern California, Los Angeles, CA, USA. [2]Department of Pathology, Keck School of Medicine, University of Southern California, Los Angeles, CA, USA. ✉e-mail: macleana@usc.edu

Hence, HSCs with weak selective advantages that arise before birth are naturally protected from stochastic loss by growth and have enough time to eventually reach detectable sizes later in life.

Studies of hematopoiesis in twins offer a unique opportunity to test whether stem cell variation arising before birth can explain CH. Approximately two-thirds of monozygotic (MZ) twins have mono-chorionic placentas and share blood circulation in utero[32,33], and therefore share HSC variation at birth. If HSC clonal variation is acquired after birth, blood clonal compositions will differ between aged MZ twins. If, on the other hand, aged MZ twins share the same clonal compositions, i.e. the same clones have grown towards fixation over decades[2,21,34], then the variation most likely arose before birth. Selection for variation before birth is supported by MZ twin studies of X-chromosome inactivation, which found a blood ratio skewing with aging, with concordance for either maternal or paternal X-chromosome dominance[35,36]. However, CH driver mutations are usually not concordant between MZ twins, indicating that driver mutations arise after birth and that MZ twins do not share a predisposition for specific mutations[34].

We hypothesize that HSC variation inevitably arises during development leading to subtle selective differences that eventually lead to CH later in life. This hypothesis is testable with MZ twins and lineage markers that become polymorphic before birth. Somatic mutations could be used to trace prenatal HSC subclones, but relatively few mutations occur in the brief time before birth and these prenatal mutations would be difficult to distinguish from postnatal mutations. An alternative to somatic mutations is DNA methylation that rapidly fluctuates between methylated and unmethylated states at certain CpG sites[37]. The higher rates of methylation fluctuation allow HSCs to become polymorphic before birth and allow lineage tracing through life. Tracking fluctuating methylation sites is a convenient lineage marker due to its much higher error rate and the broad availability of suitable blood methylation datasets.

Here, we show that fluctuating methylation patterns are often correlated in the blood between elderly MZ twins, relative to dizygotic twins or unrelated individuals. We develop a single-cell model of HSC clonal dynamics to study the origins of HSC clonal diversity, and show that variants arising before birth conferring weak selective survival advantages commonly become dominant later in life. Hence, the variation between many blood subclones that are weakly selected to grow to large sizes later in life was likely created before birth.

## Results

### Analysis of twins via fluctuating CpG methylation reveals patterns of HSC clonal dynamics over lifetime

To study HSC clonal dynamics over a human lifetime (Fig. 1A-B), we studied publicly available DNA methylation datasets (Table 1). Methylation profiles were extracted for the population of HSCs sampled for each individual in a dataset. We selected CpG sites that have a high degree of intra-individual heterogeneity, indicating that they represent flip-flopping sites, and which do not appear to be under active regulation[37]. Full details of data and data processing steps to construct methylation profiles per time point per individual are given in Methods and Supplementary Information Section S1.

We leveraged the shared prenatal circulation of monozygotic (MZ) twins to investigate the impact of stem cell variation arising during development on lifetime clonal dynamics of HSCs. We extracted methylation profiles for three groups of individuals: MZ twins, DZ twins, and unrelated individuals. The data range from time points near birth to 86 years of age. To compare average methylation profiles between individuals, we used the Pearson correlation coefficient ($R$) to assess similarity between pairs of individuals. Values of $R$ ranged from close to +1 (perfect positive correlation) to close to 0 (uncorrelated). For each set of pairs (MZ, DZ, unrelated), we calculated a line of best fit over time, as well as mean correlation values for each dataset

(Fig. 1C–F). All of the processed data used in this study are presented together in Fig. 1F.

For MZ twins (Fig. 1C, F), considerable variability was present in the correlation coefficients at birth, with some twin pairs showing much higher values than others. Variability exists in MZ twin development, whereby cleavage before formation of the blastocyst leads to dichorionic/diamniotic twins but cleavage after blastocyst formation can lead to either monochorionic or monochorionic/monoamniotic twins[38]. An estimated $\frac{2}{3}$ of MZ twins share circulation during development[39]: MZ twins with higher initial Pearson coefficients thus likely represent twins who shared circulation during development and the emergence of HSCs; MZ twins with lower initial Pearson coefficients (closer to DZ/unrelated pairs) likely characterize twins that split earlier and were thus offered less/no opportunity for shared developmental hematopoiesis. This correspondence is however currently speculative and additional data will be required for confirmation. The mean Pearson coefficients at birth were approximately 0.87 for MZ twins (Fig. 1C), 0.72 for DZ twins (Fig. 1D), and 0.61 for unrelated individuals (Fig. 1E).

Throughout the human lifespan, similar trends were observed for each group of individuals, whereby the clonal relatedness declines slowly with age. The slopes of decline differ between groups, with the sharpest decline observed for DZ twins (slope of − 0.0033), and more modest declines observed for unrelated individuals (slope of − 0.0025) and MZ twins (slope of − 0.0020). The variability at each time point was largest for MZ twins. This resulted in wide range of HSC clonal relatedness in MZ twins in later life, with Pearson coefficients ranging from low (- 0.3) to very high (> 0.9). We also noted a change in the dynamics in later life: the Pearson coefficients decrease more sharply after 65 years of age. This is consistent with previous studies that show a significant loss of clonal diversity in HSCs at this age[15].

### A single-cell resolved model of clonal hematopoiesis describes the somatic evolution of stem cells throughout life

To study the origins of clonal hematopoiesis and follow stem cell dynamics throughout life, we developed a mathematical model of methylation dynamics in HSCs, at the resolution of individual fluctuating CpG methylation (fCpG) sites in single cells (Fig. 2A; Methods). The clonal evolution of the HSC population as quantified by fluctuating methylation clocks (FMCs) was modeled from embryonic development through birth and up to a 100-year human lifespan.

At birth, a population of $N_{cell}$ stem cells represents the entire HSC pool, and we assume a constant population size throughout life (Fig. 2B). Variation in the population size during life does occur, e.g. the HSC expands during aging, however these changes are small relative to both the growth phase during development and the total population size after birth. Each cell is modeled by $N_{site}$ fCpG sites (Fig. 2A), where at each site, given two alleles, there are three possible methylation states: 0 (unmethylated), 0.5 (hemimethylated) and 1 (fully methylated). A population of cells is defined by $\mathbf{x}$, $\mathbf{x} = [x_i^j]$, where $x_i^j$ represents the methylation status of the $i^{th}$ cell at the $j^{th}$ fCpG site, i.e. we have $x_i^j \in \{0, 1, 2\}$, for $i \in \{1, 2, ...N_{cells}\}$ and $j \in \{1, 2, ...N_{sites}\}$. We model stem cell methylation dynamics throughout life by a Markov process. Dynamics occur by cell replacement (with probability $\alpha$) and changes in fCpG methylation. During a cell replacement event, each fCpG site can flip with probability $\gamma$, and a methylation step change of ± 1. See Table 2 for full description of model parameters along with values used for simulation.

In development, hematopoiesis was modeled starting from a small initial number of HSCs. This population of initial stem cells (of size $N_{clone}$) each seeds a clone that grows during development until the population reaches its final size of $N_{cell}$ cells (Fig. 2B). Each fCpG site in each clone is initialized randomly in state either 0 (unmethylated) or 1 (fully methylated). To model selection in development and throughout life, we define that each of the $N_{clone}$ clones has a fitness coefficient,

1 + $s_i$, for $i \in \{1, 2, ..., N_{clone}\}$, with $s_i$ chosen from a Gamma distribution (see Supplementary Table S1).

The model (Fig. 2A) is developed to follow single-site single-cell methylation dynamics. At the population level, it is similar to previous FMCs models that allow for analysis of average methylation profiles in a population of cells[15,20,37]. In particular, for increasing number of clones and/or increasing rate of flipping, we see the average methylation profile switch from a multi-modal distribution to a unimodal

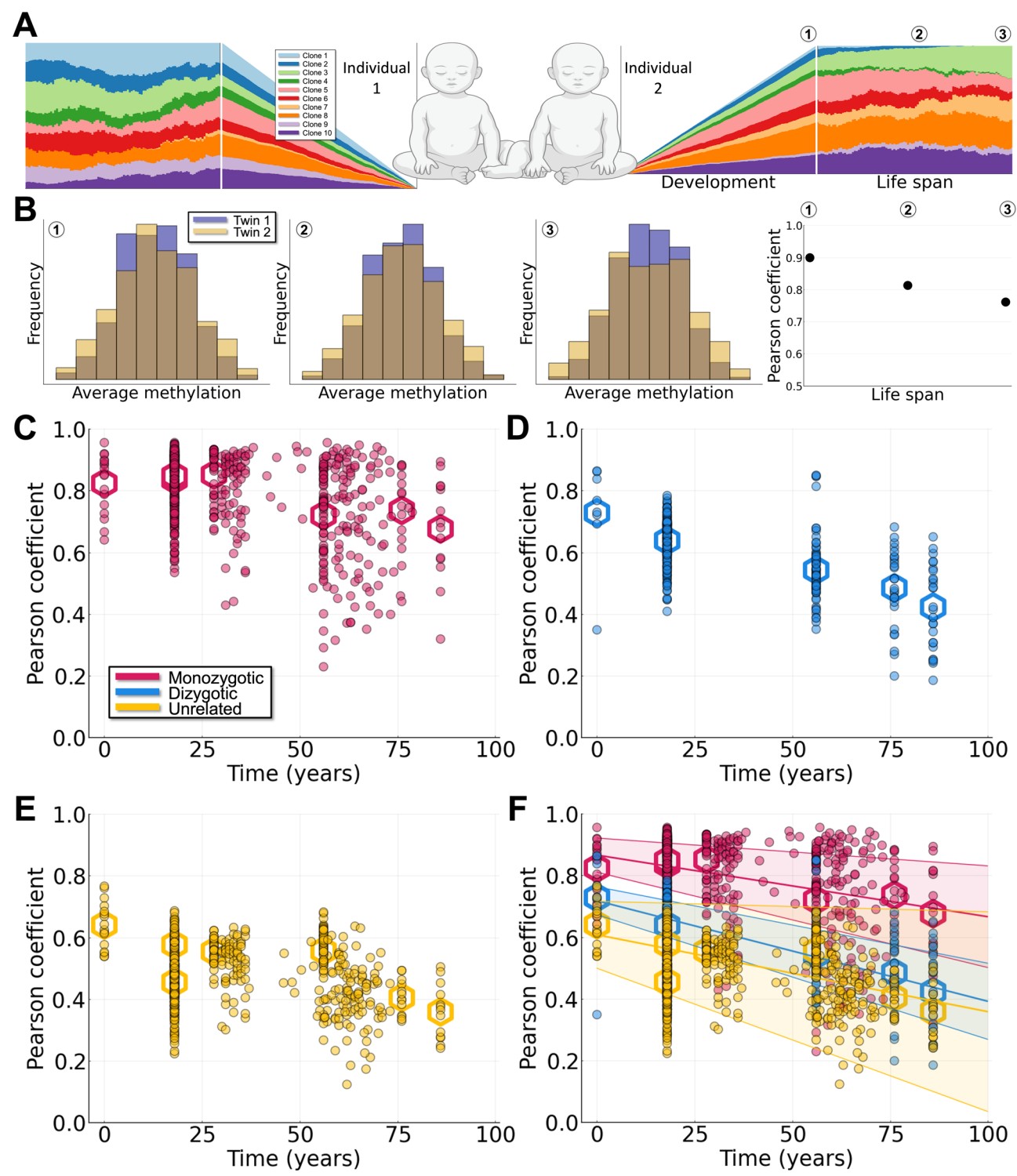

**Fig. 1 | Clonal dynamics of HSCs are characterized by methylation profiles over lifetime. A** Cartoon depicting HSC population dynamics from embryo development throughout a human lifespan for two twins. Colors represent different HSC clones. Created in BioRender. Kreger, J. (2024) BioRender.com/m1o751. **B** β distributions of two twins over time, see circled numbers in panel A. Colors represent the different individuals. The Pearson correlation coefficient between individuals over time is also shown (right panel). **C** Pearson correlation coefficients for average methylation profiles between MZ twins. Dots represent individual comparisons and hexagons represent means of datasets. **D** Pearson correlation coefficients between DZ twins. **E** Pearson correlation coefficients between unrelated individuals. **F** Pearson correlation coefficients for all of the three comparisons (compilation of the data presented in **C**–**E**), along with lines of best fit and corresponding 90% confidence intervals. The lines of best fit and confidence intervals are calculated using the means of datasets (hexagons).

**Table 1 | Description of datasets**

| GEO ID | Reference | Participant age (years) | Num. mono-zygotic twin pairs | Num. dizy-gotic twin pairs |
|---|---|---|---|---|
| GSE36642 | 68 | Prenatal (gestational week 32-38) | 17 | 8 |
| GSE154566 | 69 | 18 | 116 | – |
| GSE105018 | 70 | 18 | 426 | 306 |
| GSE43975 | 71 | 28 (average) | 39 | – |
| GSE61496 | 72 | 30–74 | 142 | – |
| GSE89093 | 73 | 38–79 | 46 | – |
| GSE100227 | 74 | 56 (average) | 65 | 66 |
| GSE73115 | 75 | 76 (average) | 28 | 52 |
| GSE73115 | 75 | 86 (average) | 28 | 52 |

The first column is the dataset GEO ID, the second column is the reference, the third column is the age of the individuals, the fourth column is the number of MZ twin pairs, and the fifth column is the number of DZ twin pairs.

distribution centered around 0.5[37]. For more details on population level similarities to previous models see Supplementary Information Section S2.

We characterized the effects of each model event: cell replacement (with rate $\alpha$; Fig. S1) and (de)methylation (with rate $\gamma$; Fig. S2) by exploring the parameter space of these parameters (Section S3 in the Supplementary Information). We also analyzed the impact of population-level model parameters on the lifetime stem cell dynamics, namely by varying the threshold number of cells at which embryos split, $N_{split}$ (Fig. 2C–E), or by varying the total number of cell clones, $N_{clones}$ (see Section S4 and Figs. S3 and S4 in the Supplementary Information). As the total number of cell clones is reduced, so is the total observed variability in the clonal distributions, resulting in overall higher Pearson coefficients between individuals (Fig. S3).

## Stem cell variation arising during development explains the blood clonal composition at birth

During development, a nascent population of HSCs must grow large enough to entirely support hematopoiesis by birth. We model a population of $N_{clone}$ cells that grows to a size of $N_{cell}$ cells, by either uniform or frequency-dependent growth[22,40,41] (Fig 1A). The variation observed between pairs of twins/unrelated individuals at birth (Fig. 1F) amounts to Pearson coefficients observed ranging from 0.5 to 1.0. In the case of MZ twins, we initialize a shared set of clones and uniform growth of these clones was not able to explain the data (Fig. 3A). The methylation distributions observed under the uniform growth model were highly similar with Pearson coefficients ≥0.95 (Fig. 3A). Positive frequency-dependent growth, in contrast, permitted greater variation in methylation profiles between individuals at birth, due to more divergent HSC clonal dynamics in development (Fig. 3B). While the positive frequency-dependent growth model (where clones with high frequency are more likely to be chosen for reproduction) is simplistic with regards to the evolutionary dynamics, it provides parsimonious means with which to generate diverse clonal distributions as observed in the data for pairs of individuals at birth.

Under the frequency-dependent growth model, we model differences in development between twins and unrelated individuals as follows. In the case of twins, we model the embryo growing from $N_{clone}$ cells to $N_{split}$ cells with identical clonal dynamics, at which point the embryos split and two independent embryos are simulated where, in each, the HSC population grows from $N_{split}$ cells to its full size of $N_{cells}$ cells. The assumptions required for this model are that: 1. MZ twins that develop from a single embryo can directly share early hematopoietic cells for a varying length of time depending on their developmental state, ranging from dichorionic/diamniotic to

monochorionic/monoamniotic, with a corresponding variation in the extent to which early hematopoietic cells may be shared; 2. due to genetic similarity, DZ twins share some clonal growth characteristics during early development; 3. unrelated individuals that share no developmental history are seeded by HSC populations that each grow independently. The data characterizing relatedness of HSCs at birth between individuals (time point 0 in Fig. 1F) are consistent with this model.

To determine values for $N_{split}$ for DZ and MZ twins, we fit the model parameters ($N_{clones}$, $N_{split}$) to FMC data characterizing twins and unrelated individuals (Fig. 2C). We fit curves to characterize the observed Pearson coefficients as we vary the number of shared cells, and find that, assuming $N_{clones} = 10$, the following values of embryo splitting are consistent with the data: for MZ twins $N_{split} \approx 36$; for DZ twins $N_{split} \approx 15$; and for unrelated individuals $N_{split} = 10$ (i.e. no shared development). For further details, see Supplementary Information Section S5 and Figs. S5 and S6.

## HSC clonal dynamics in monozygotic twins are constrained by weak selection

In order to explore the effects of selection on clonal dynamics of HSCs, we simulated FMC dynamics in HSCs over a human lifetime under different models for selection. We considered three models: no selection, weak selection, and strong selection. In Fig. 4A–C, simulations of the lifetime HSC dynamics for MZ twins are shown, under each of three different selection models. In each panel, the full dataset is plotted (see Fig. 1F) for comparison against different model simulations. In the case of no selection, where all clones have the same fitness post-birth (clonal dynamics during development are subject to frequency-dependent selection), Pearson coefficients later in life are too low, as clonal distributions diverge between individuals (Fig. 4A for a representative simulation; and Fig. 4D, for trajectories of 100 pairs of individuals). In this regime characterized by drift, many twin pair methylation profiles become under-correlated over time relative to the data.

For non-neutral models of lifetime HSC dynamics, stem cell fitness values were drawn from a Gamma distribution parameterized by shape $a$ and size $\theta$ (see Methods for details). For weak selection $a = 0.05$ and $\theta = 0.01$ (see Table S1). Variants with weak fitness advantages arising in development can experience clonal expansions due to the developmental growth phase (Fig. 4B and E). Even if the variant frequency is low at birth, the long time range of a human lifespan enables the effects of weak selection to become evident in later life. In simulations, we observed that many MZ twin pairs remain correlated under weak selection throughout life, but without individual clones fixating. We observe that weak selection results in good agreement between our model simulations and the MZ twin data. In particular, through directly comparing model simulations to the data (see the last section in Methods and Table S2 in the Supplementary Information) we find that weak selection results in a better fit to the data compared to neutral selection, as measured by the mean distance from each model simulation to simulated data trajectories.

For strong selection, clone fitness values were sampled from a Gamma distribution with $a = 0.05$ and $\theta = 0.05$ (see Table S1). In this case we observed frequent, dramatic reductions in clonal diversity whereby one clone with relatively high fitness would rapidly expand and fixate in the HSC population (Fig. 4C, F). This scenario mimics the possible endpoint following from trajectories of clonal hematopoiesis of indeterminate potential (CHIP)/hematopoietic malignancies in the sense that all clonal diversity has been lost. When the same clone nears/reaches fixation in two people due to its fitness, the Pearson coefficient between individuals approaches 1. In this scenario, the methylation distribution in both individuals becomes bimodal with most density near 0 and 1. The bimodal distribution (rather than tri-modal or "W-shaped" as in Ref. 37) observed in each twin when a clone

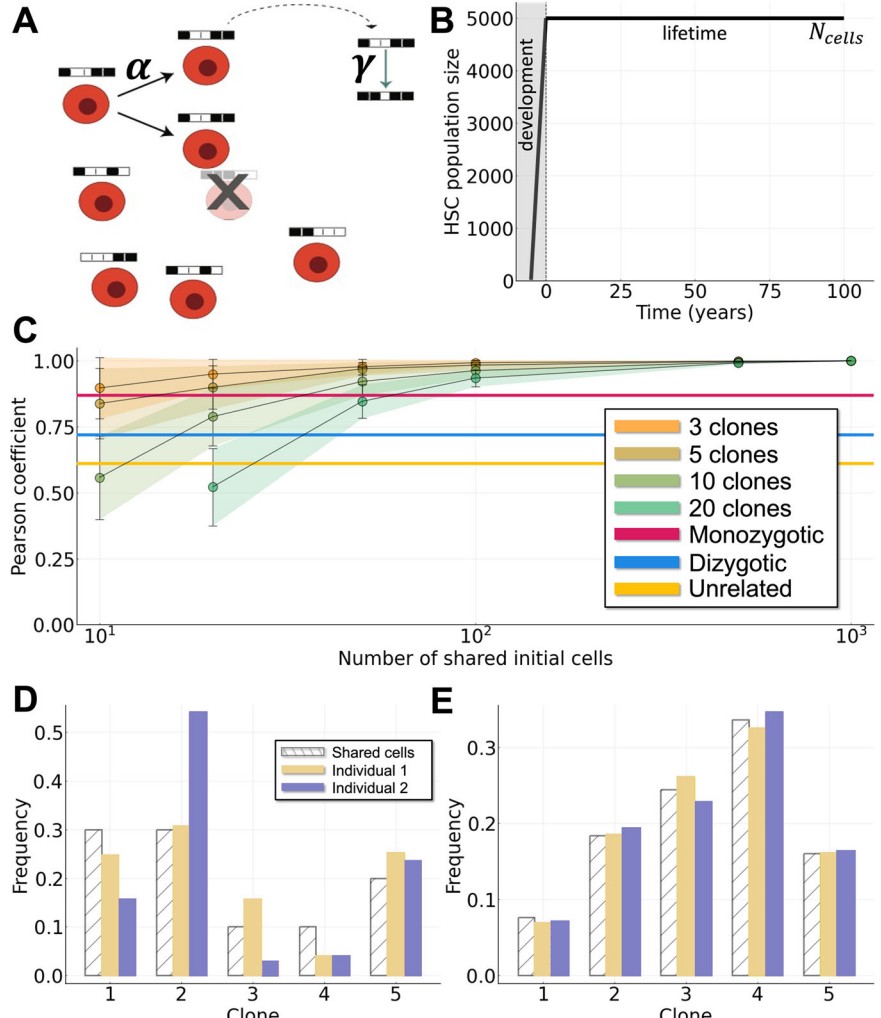

**Fig. 2 | Variation during development is needed in order for model to match initial Pearson coefficients at birth. A** Schematic diagram of the mathematical model, with probability of cell replacement $\alpha$ and probability of change in methylation $\gamma$. **B** HSC population dynamics over a lifetime. **C** Parameter estimation using initial Pearson coefficients of twins and unrelated individuals. Given a choice for the initial number of clones, the number of cells at which the embryo splits ($N_{split}$) can be determined. Data points (dots) represent the mean Pearson coefficient for $10^3$ independent simulations of the model with a given number of clones

($N_{clones}$) and number of shared cells ($N_{split}$). Shaded areas and error bars denote one standard deviation from the mean over the $10^3$ simulations. The horizontal lines represent the data: initial Pearson coefficients at birth for MZ twins (red), DZ twins (blue) and unrelated individuals (yellow). **D**–**E** Effect of varying $N_{split}$. White striped bars represent the clonal distribution in the shared embryo when it reaches $N_{split}$ cells and splits into two embryos. Yellow/blue bars represent individuals 1 or 2 at the point at which the embryo has reached $N_{cells}$ cells and finished growing. D: $N_{split} = 10$. E: $N_{split} = 500$. All other parameter values can be found in Table 2.

fixates (Fig. 4C) results from the large population size of HSCs ($5 \times 10^3$ cells) and the relatively low CpG flipping rate.

We also observed infrequently scenarios where two or more clones have relatively high fitness, and then due to differences arising from frequency-dependent growth during development different clones will tend towards fixation in the two twins. This results in a Pearson coefficient near or trending toward 0 (Fig. 4F). Moreover, we observed in the case of strong selection that the clonal distributions at birth had little effect on the lifetime clonal dynamics, as long as the fittest clones were not lost due to random cell loss. This is due to the relative fitness advantages of these clones and the length of the possible human lifetime: even if the fittest clone starts with few initial cells at birth relative others, over many decades it will grow to dominate, even in the large size of the HSC pool (the classical probability of fixation in a large population for a single cell with fitness advantage $s$ is given by $2s$[18]). The timing of when variants arise under strong selection does not greatly affect HSC clonal dynamics in later life since a loss of clonal diversity is effectively ensured. Under this model, the timing of when variants arise would affect only the time at which the clone takes

over (earlier generation/faster initial growth resulting in earlier fixation).

## Different lifetime clonal dynamics of twins vs unrelated individuals can be explained with no tunable parameters other than $n_{split}$

In the previous section we studied the clonal dynamics of HSCs over lifetime in MZ twins, under different models of selection. We analyzed simulations of DZ twins and unrelated individuals in a similar manner and found evidence that in both cases strong selection is not consistent with the data (see Supplementary Information Section S6 and Figs. S7–S9). In the case of DZ twins/unrelated individuals, lifetime FMC dynamics may be consistent with either neutral dynamics or weak selection (Table S2), although as we have seen neutral dynamics are not entirely consistent with clonal dynamics in MZ twins.

We simulated 100 pairs of individuals from each group (MZ twins, DZ twins, and unrelated individuals) over a human lifetime under the assumptions of 1) frequency-dependent growth during development and 2) weak selection throughout life (Fig. 5). All parameters of the

**Table 2 | Description of model parameters and values**

| Notation | Description | Value | Units | Reference |
|---|---|---|---|---|
| $N_{sites}$ | Number of fCpG sites per cell | $10^3$ | – | – |
| $N_{cells}$ | Number of cells | $5 \times 10^3$ | Cells | – |
| $N_{clones}$ | Number of cell clones | 10, varies | Cells | – |
| $N_{split}$ | Cell threshold at which the embryos split | 10, 15, 36 | Cells | – |
| $\alpha$ | Cell replacement (birth/death) rate | $\frac{1}{365}$, varies | Days$^{-1}$ | 15,30 |
| $\gamma$ | (de)methylation rate | $10^{-3}$, varies | Per fCpG site per replacement event | 37,76,77 |
| $1 + s_i$ | Fitness coefficient for $i^{th}$ clone | Table S1 | – | 20 |
| $a$ | Gamma distribution shape parameter | 0.05 | – | 63–65 |
| $\theta$ | Gamma distribution scale parameter | 0.01, 0.05 | – | 63–65 |

Parameter values are estimated from the literature where possible, see in particular[15,37]. The first column is the parameter notation, the second column is the parameter description, the third column is the parameter estimated value, the fourth column is the parameter units (if applicable), and the fifth column is the citation of the reference for the parameter estimate.

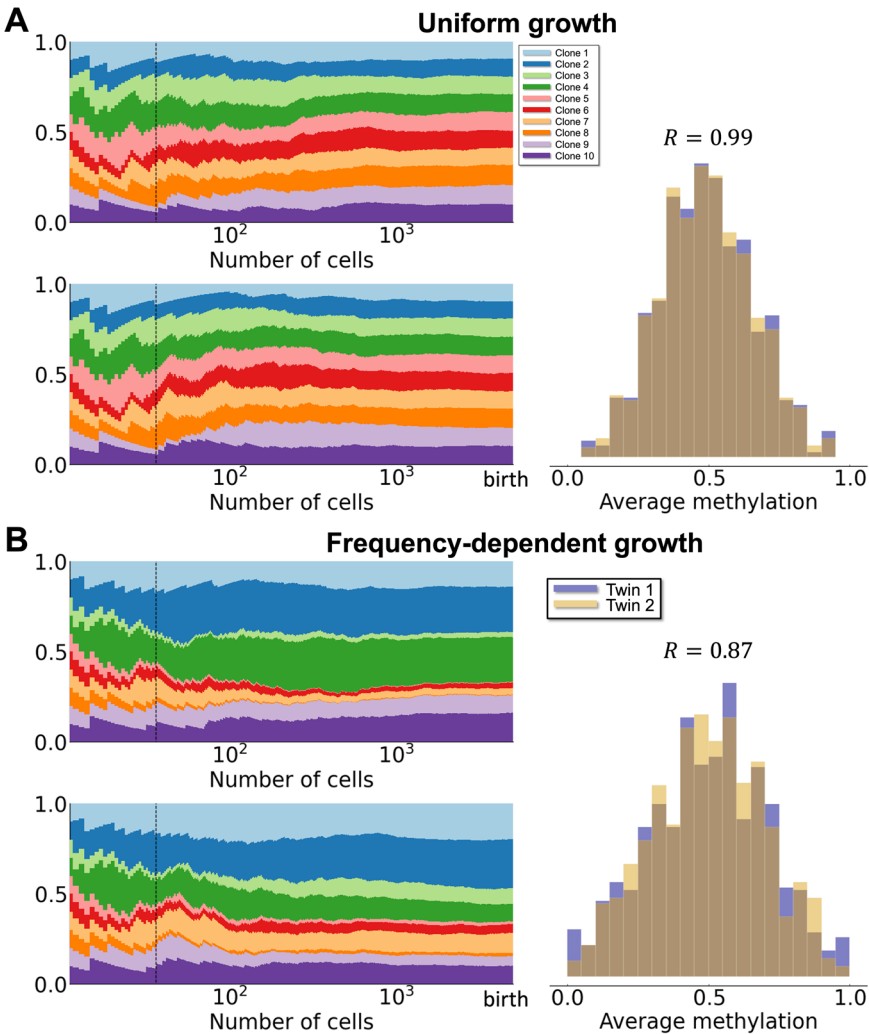

**Fig. 3 | Frequency-dependent growth produces clonal variation during development. A** Uniform growth during development with $N_{split} = 36$ and no selection ($s_i = 0$) for all clones. Left panel: Clonal dynamics during development, colors denote different clones. Right panel: $\beta$ distributions at the end of development, i.e.

birth. The initial Pearson correlation coefficient at birth is 0.99. **B** Same as A for frequency-dependent growth model during development. The initial Pearson correlation coefficient at birth is 0.87. All other parameter values can be found in Table 2.

model were held constant over all pairs of individuals except for $N_{split}$, which is determined from twin status: $N_{split} = 36$ for MZ twins (Fig. 5A); $N_{split} = 15$ for DZ twins (Fig. 5B); and $N_{split} = 10$ for unrelated individuals (Fig. 5C). We observed that the balance between clonal diversity arising from frequency-dependent growth and weak selection throughout life

gave rise to more similar (MZ) or divergent (unrelated) methylation distributions in later life (Fig. 5A–C, right hand column). Various methylation distributions can be observed in cases with reduced clonal diversity, leading to multimodal distributions with three modes if two clones dominate (Fig. 5C; individual 2), or more if 3–4 clones dominate

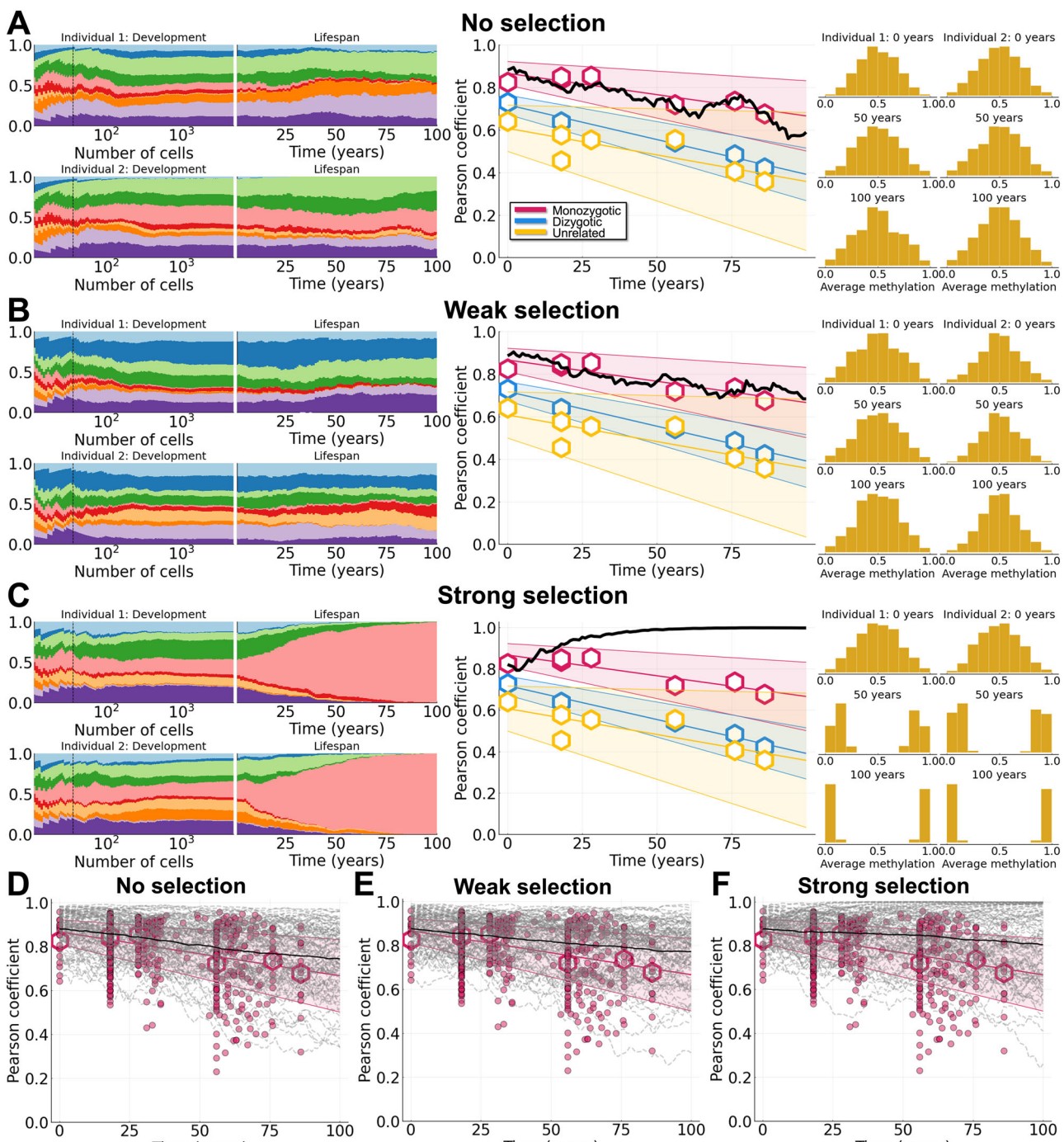

**Fig. 4 | Variants with weak selection arise during development and explain FMC dynamics for monozygotic twins.** Simulations shown for MZ twins ($N_{split} = 36$) with frequency-dependent growth during development. For plots of the data (middle column and bottom row): dots represent individual comparisons, hexagons represent means of datasets, and shaded bands represent 90% confidence intervals. The data plotted in each panel (A-C and D-F) are the same for purposes of comparison against different model simulations. **A–C** Clone growth frequency plots for both individuals during development and life (dashed vertical line represents $N_{split}$), Pearson correlation coefficient, and $\beta$ distributions at 0, 50, and 100 years of life. **A**: No selection. **B**: Weak selection ($a = 0.05$ and $\theta = 0.01$). **C**: Strong selection ($a = 0.05$ and $\theta = 0.05$). **D–F** Results from $10^2$ simulations are shown (dashed lines are individual simulations and solid lines are mean trajectories). **D**: No selection. **E**: Weak selection ($a = 0.05$ and $\theta = 0.01$). **F**: Strong selection ($a = 0.05$ and $\theta = 0.05$). All other parameter values can be found in Table 2.

(Fig. 5C; individual 1). For all of the total of 100 simulations for each twin status (Fig. 5D–F, dashed lines), we observed a good fit between model and data in each of the three twin cases modeled. We considered an alternative selection model in which clones initially evolved neutrally but in which over time fitness coefficients grow in magnitude based on their respective frequencies (see Supplementary Information Section S7 and Fig. S10). We found this scenario was more likely to lead

to scenarios in which driver mutations will cause large clonal expansions and take over the HSC niche of an individual; in this regard it is similar to the strong selection regime modeled, and does not match what is observed in the data. Finally, we also analyzed $\beta$-distributions of individuals with a clinical CHIP diagnosis (see Supplementary Information Section S8 and Fig. S11). Here we observe an increased variation in the $\beta$-distribution due to the loss of clonal diversity, which will

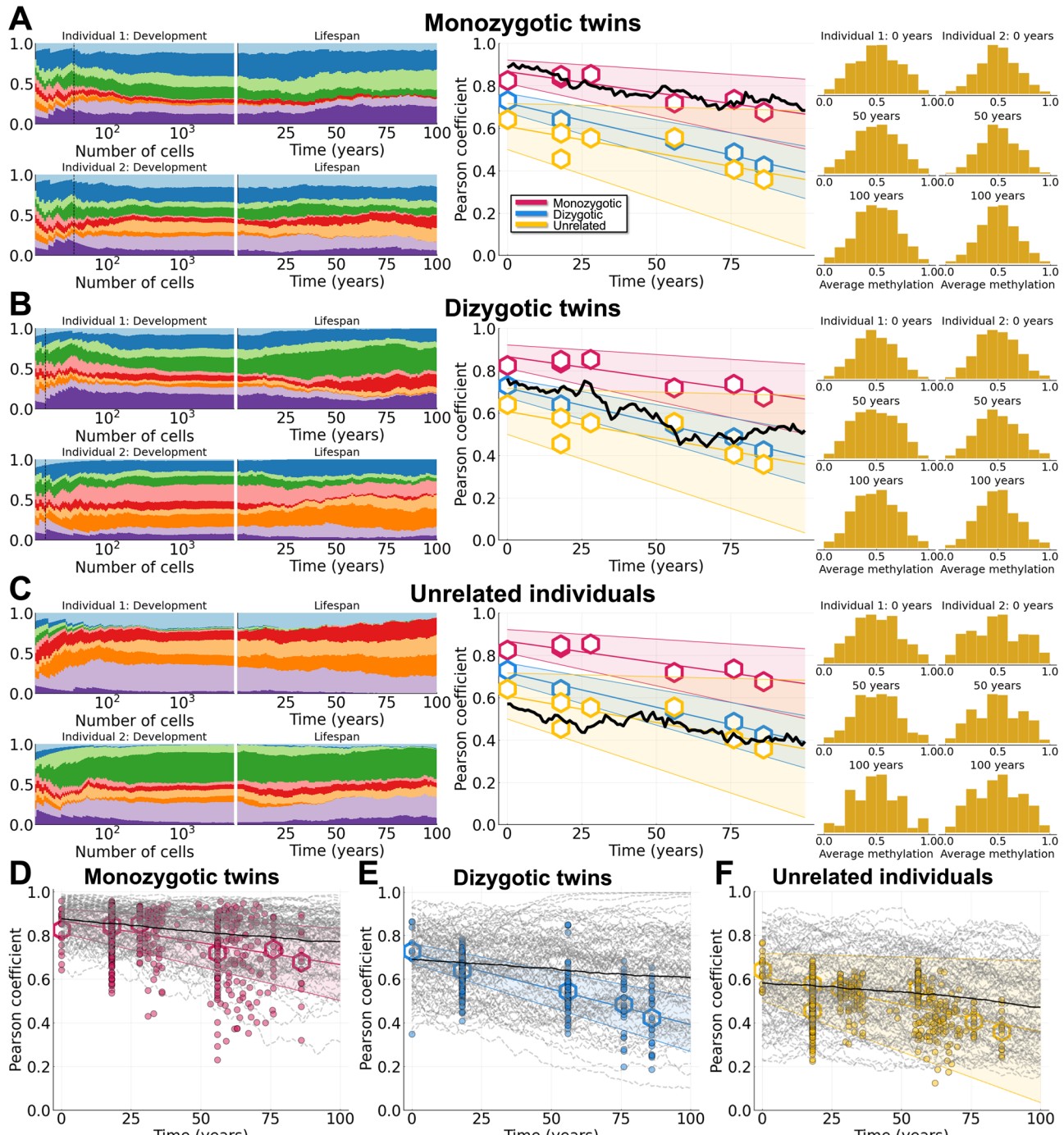

**Fig. 5 | Weak selection with variants arising in development explains lifetime dynamics of twins and unrelated individuals.** Simulations shown for weak selection ($a = 0.05$ and $\theta = 0.01$) with frequency-dependent growth during development. For plots of the data (middle column and bottom row): dots represent individual comparisons, hexagons represent means of datasets, and shaded bands represent 90% confidence intervals. The data plotted in each panel (A-C and D-F) are the same for purposes of comparison against different model simulations. **A–C** Clone growth frequency plots for both individuals during development and life (dashed vertical line represents $N_{split}$), Pearson correlation coefficient, and $\beta$ distributions at 0, 50, and 100 years of life. **A:** MZ twins, $N_{split} = 36$. **B:** DZ twins, $N_{split} = 15$. **C:** Unrelated individuals, $N_{split} = 10$. **D–F:** Results from $10^2$ simulations are shown (dashed lines are individual simulations and solid lines are mean trajectories). **D:** MZ twins. **E:** DZ twins. **F:** Unrelated individuals. All other parameter values can be found in Table 2.

increase even more if more rapid clonal expansions occur (e.g. acute leukemias, major hematopoietic neoplasms)[37].

In summary, when variants subject to weak selection arise during development, they oftentimes experience clonal expansion because of growth, which mitigates against stochastic loss. Then, post-birth, over the relatively long possible human lifespan, the effects of the weak fitness advantages of particular clones can become evident. By

studying the lifetime dynamics of HSCs in twins and unrelated individuals, we show how variation is most likely generated before birth yet only becomes evident in later life.

## Discussion

Clonal hematopoiesis becomes increasingly detectable with aging. In many cases, driver mutations are absent and inferred weak selection

indicate variants arise much earlier in life with unexplained initial growth spurts[15]. Here, we have shown that a simple model of HSC clonal dynamics with weak fitness advantages can explain lifetime hematopoiesis when variants arise before birth.

Publicly available twin methylation data offer a unique opportunity to study when hematopoietic variants arise. Through theory and simulation, we have shown that weakly selective variants present in embryos can undergo expansion as a byproduct of developmental growth, which mitigates against random loss, and can permit clones to tend towards fixation much later in life after a period of latency. Analysis of the clonal dynamics of twins vs unrelated individuals showed that some HSC variation is required to occur before birth. Given the assumption that some HSC clonal variation is present by birth, we are able to explain the different lifetime trajectories of pairs of unrelated vs twinned individuals using a single parameter that characterizes the developmental growth phase of hematopoiesis. With no further fitting, this model is consistent with the increased frequency of clonal expansions observed in individuals over the age of 65, without the need for any driver mutations to arise later in life.

The fate of genetic variants within a population has a rich theoretical foundation[18,19,23,24], although in cases of time-varying population sizes, these inferences are more nuanced[28,29,42]. A growing body of literature has revealed in depth the clonal diversity and clonal dynamics of HSCs over life[4–7,15,16]. Evidence from these studies suggests that HSC variation is likely to arise early, perhaps even before birth. However, sampling hematopoiesis before birth for longitudinal study is impractical. By overcoming this challenge using twins data from which we can estimate variation occurring before vs after birth, we are able to show that indeed the developmental phase of hematopoiesis offers a window of opportunity for genetic variants to persist even when they have only very slight fitness advantages. In contrast, previous models assumed higher selection coefficients to explain clonal distributions in individuals ≥65 years old[15], which were necessary if these variants arose in mid life, and not earlier. In our model, low frequency non-driver mutations are "passengers" that hitchhike with the selected prenatal HSC subclones.

Genetic heterogeneity of HSCs is but one factor influencing hematopoiesis as we age, since HSCs are also a product of their niche, and the stem cell niche/microenvironment plays a crucial role in defining stem cell function[43]. Additional non-genetic factors influencing hematopoiesis include stem cell heterogeneity (mediated by transcriptional or epigenetic variation)[44–47], feedbacks from and contributions of progenitor cells to hematopoiesis[48] and other environmental signals, e.g. resulting from diet or the immune system[49,50]. Germline genetic variation can also alter risk of both clonal hematopoiesis and myeloid malignancies[51–53]. Future models building on this work ought to consider some of these factors and their influence on healthy hematopoiesis in development and throughout life.

Limitations of the current model include the scaled HSC population size (around 0.1–0.2 of the estimated number of human HSCs[31]), the assumption of constant selection throughout life, and the simplicity of the model used for hematopoietic development in utero. Future work could include further biological details[54] within the mathematical framework considered here in order to analyze their effects and increase the predictive power of the model[15,55,56]. This could include time-dependent non-constant fitness effects, negative frequency-depedent selection[57], as well as partially shared bone marrow microenvironments for twins (vs unrelated individuals). It would also be interesting to analyze how individuals with confirmed clonal hematopoiesis behave with respect to fCpG methylation patterns and if the clonal fractions estimated by methylation dynamics corresponds with the sizes of genetic clones.

In summary, we have shown that HSC variants created before birth can determine HSC clonality much later in life. Fluctuating CpG methylation provides a lineage marker sufficient to reveal clonal variation in development and early life. By modeling this variation, we have seen that weak selection conferred by the stem cell variation that arose during development yields clonal hematopoiesis decades later. Although CH is associated with increased disease risks, HSC variants created before birth combined with decades of selection within a large HSC pool can help explain why CH is so common with normal aging. Further more detailed studies of MZ twin blood populations, especially elderly MZ twins with documented monochorionic or dichorionic placentas, can help further explore the roles of early HSC variation on normal aging and disease predisposition.

## Methods

### Data and fCpG site filtering
Methylation datasets are publicly available on the Gene Expression Omnibus (GEO)[58]. For this study, we use data listed in Table 1, which represent a collection of DNA methylation profiles from both monozygotic (MZ) and dizygotic (DZ) twins. We extract average methylation profiles from groups of MZ and DZ twins at different points throughout the human lifespan (Fig. 1). Comparisons of unrelated individuals are made by randomly sampling pairs of individuals that are unrelated from the twins datasets. Similar to[37], we study the dynamics of "neutral" fCpG sites, i.e. CpG loci that are not actively regulated and that show a high degree of intraindividaul heterogeneity. In particular, we curate a list of HSC-specific CpG loci that are not associated with any known single nucleotide polymorphisms, see Supplementary Information for further details. To compare the average methylation profiles between individuals we use the Pearson correlation coefficient. For each category (MZ twins/DZ twins/unrelated individuals) we calculate the line of best fit for the data as well as the 90% confidence interval using the LsqFit.jl Julia curve fitting package[59].

All raw data (DNA methylation profiles) used in this study are publicly available on the Gene Expression Omnibus (GEO) (see above for GSE numbers and citations). All processed data (Pearson correlation coefficients between twin/unrelated individuals at different ages) is publicly available here: github.com/maclean-lab/scFMC-model. Data used in this study can also be seen in Fig. 1C-F. A full list of "neutral" fCpG sites used in this study can also be found at our publicly available github repository: github.com/maclean-lab/scFMC-model, see also Supplementary Information.

### Correlations between individuals
In order to compare the correlation between average methylation profiles at different ages we use the Pearson correlation coefficient[60], which is given by

$$r = \frac{\sum (x_i - \bar{x})(y_i - \bar{y})}{\sqrt{\sum (x_i - \bar{x})^2 (y_i - \bar{y})^2}}, \tag{1}$$

where $x_i (y_i)$ is the average methylation value of individual 1 (2) at the $i^{th}$ fCpG site and $\bar{x} (\bar{y})$ is the mean value for individual 1 (2) over all fCpG sites. The Pearson coefficient measures the similarity between the two average methylation profiles, where a Pearson coefficient of 1 represents a positive correlation, a Pearson coefficient of $-1$ represents a negative correlation, and a Pearson coefficient of 0 represents no relationship.

### Model description: modeling HSC dynamics postnatally
We develop a theoretical model of methylation dynamics in a population of cells, similar to[37]. We model dynamics at the resolution of individual fCpG sites in individual hematopoietic stem cells (HSCs). We assume there are $N_{cells}$ cells, each with $N_{sites}$ fCpG sites. Since there are two alleles at each site, there are three possible methylation states at each site

- 0 (which represents 0% methylation),
- 1 (which represents 50% methylation),
- 2 (which represents 100% methlyation).

This can be seen in Fig. 2A (for a similar representation see Fig. 1 in[37]). We represent **x** as our population of cells, where

$$
\mathbf{x} = \begin{bmatrix}
x_1^1 & x_2^1 & \ldots & x_i^1 & \ldots & x_{N_{cells}}^1 \\
x_1^2 & x_2^2 & \ldots & x_i^2 & \ldots & x_{N_{cells}}^2 \\
\ldots & \ldots & \ldots & \ldots & \ldots & \ldots \\
x_1^j & x_2^j & \ldots & x_i^j & \ldots & x_{N_{cells}}^j \\
\ldots & \ldots & \ldots & \ldots & \ldots & \ldots \\
x_1^{N_{sites}} & x_2^{N_{sites}} & \ldots & x_i^{N_{sites}} & \ldots & x_{N_{cells}}^{N_{sites}}
\end{bmatrix},
$$

and each $x_i^j \in \{0, 1, 2\}$ represents the methylation status of the $i^{th}$ cell at the $j^{th}$ fCpG site for $i \in \{1, 2, ...N_{cells}\}$ and $j \in \{1, 2, ...N_{CpG}\}$. We develop a discrete Markovian model where each discrete step represents one day, and we simulate the model up to 100 years (assuming 365 days a year). For post-birth dynamics (embryo development described in the following paragraph), we assume a constant population of cells. Dynamics occur with cell replacement (a birth/death event), which happens for each cell with probability $\alpha$ each discrete step. We assume the cell chosen to be replaced (die) is chosen uniformly from the population of cells, and the cell chosen to reproduce is chosen based on fitness (described below). In this way, a constant population size of HSCs is maintained from birth until 100 years of age. Furthermore, during each replacement event, each fCpG site in the new cell has a $\gamma$ chance to flip, with maximum methylation step $\pm 1$ (Fig. 2A).

## Model description: modeling HSC dynamics during development

To model development (before birth), we assume there are $N_{clones}$ cell clones and we start initially with one cell of each clone. Each fCpG site for each clone is initially randomly either 0 (homozygously unmethylated) or 2 (homozygously methylated). This is because during the early stages of embryogenesis, the inherited methylation patterns from parental gametes are largely erased before a large wave of de novo methylation remodels the entire genome, resulting in a bimodal methylation distribution[37,61]. To model selection, we assume that each of the $N_{clones}$ clones has a fitness coefficient, $1 + s_i$, for $i \in \{1, 2, ..., N_{clones}\}$, where each $s_i$ is chosen from a Gamma distribution[62] with shape parameter $a$ and scale parameter $\theta$[63–65]. For MZ/DZ twins the clonal fitness coefficients are the same for both individuals across individual comparisons whereas for unrelated individuals the clonal fitness coefficients are different (but chosen from the same distribution) for both individuals across individual comparisons. (De)methylation at fCpG sites will change the methylation profile of cells within a clonal lineage, but we assume that this has a negligible effect on fitness. We note that MZ twin dynamics over lifetime (as characterized by methylation profiles) match the case of no selection as the fitness differences go to zero, i.e. as clones have fitness coefficients $s_i = 0 \ \forall \ i = 1, 2, ..., N_{clones}$).

$$
\lim_{a \to 0 \text{ or } \theta \to 0} \text{clonal distributions} = \text{clonal distributions}_{s_i = c, \text{ for all } i}. \quad (2)
$$

We allow the $N_{clones}$ cells to then grow to $N_{cells}$ cells (see Fig. 2B) in either of two different growth scenarios-

- **No/little variation during development**: uniform growth, where each clone has an equal chance throughout developmental growth to reproduce. Here, each embryo will have frequencies of approximately $\frac{1}{N_{clones}}$ for each of the $N_{clones}$ clones.
- **Variation during development**: positive frequency-dependent growth, where clones that grow to larger numbers during early development are more likely to reach higher frequencies in the

embryo[41]. Here, we also assume growth which is weighted by each clone's fitness coefficient $(1 + s_i)$. This increases the variation in the clonal frequencies during development and at birth.

In the case of MZ and DZ twins, which share a single embryo early during development (and blood circulation in the case of MZ twins), we allow a single embryo to grow from $N_{clones}$ cells to $N_{split}$ cells before splitting into two embryos. These two embryos then grow independently to the full $N_{cells}$ cells. The value of $N_{split}$ is determined based on simulating embryo development and finding the best fit to Pearson coefficients at birth from twins from methylation data (Fig. 2C). Larger values of $N_{split}$ result in less variability between individuals (Fig. 2D-E). For unrelated individuals, we assume that $N_{split} = N_{clones}$ (i.e. there is no shared embryo growth).

All models were developed in the Julia programming language[66,67]. All code developed for this study is available at a public github repository located here: github.com/maclean-lab/scFMC-model.

## Parameter estimates

Model parameters are chosen based on previous literature estimates and/or best fit to available data. For a full list of model parameters and their values, see Table 2. For the number of cells ($N_{cells}$) and number of fCpG sites per cell ($N_{sites}$), we choose values that are reasonable from both a biological and computational standpoint. Examples of fitness coefficients $(1 + s_i)$ for each clone in the case of strong, weak, and no selection are included in Table S1 in the Supplementary Information.

## Comparing model simulations vs methylation data

To directly compare model simulations to the methylation data we generate $10^2$ data trajectories for monzygotic twins, dizygotic twins, and unrelated individuals, each in the case of no selection, weak selection, and strong selection during life. Simulated data trajectory points are generated for each dataset (e.g. at $t = 0, 18, 28, 56, 76, 86$ for MZ twins) via a normal distribution with mean of the dataset and standard deviation of the dataset at that time point. These trajectories are then compared with the $10^2$ mathematical model simulations by computing the function $d$, which measures the mean distance from each model simulation to each data trajectory at the given time points. Results can be seen in Table S2 in the Supplementary Information.

## Reporting summary

Further information on research design is available in the Nature Portfolio Reporting Summary linked to this article.

## Data availability

All code and data analysis associated with this study are available on GitHub at: github.com/maclean-lab/scFMC-model. All raw datasets used in this study are publicly available on the Gene Expression Omnibus (GEO): see Table 1. Source data are provided with this paper.

## Code availability

All code and data analysis associated with this study are available on GitHub at: github.com/maclean-lab/scFMC-model. All raw datasets used in this study are publicly available on the Gene Expression Omnibus (GEO): see Table 1.

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

## Acknowledgements
J.K. acknowledges support from the Momental Foundation Mistletoe Research Fellowship. A.L.M. acknowledges support from the NIH (R35GM143019).

## Author contributions
J.K.: Conceptualization, software, investigation, methodology, writing-original draft, and editing. J.A.M.: Investigation, methodology, writing-original draft, and editing. D.S.: Conceptualization, investigation, methodology, supervision, writing-original draft, and editing. A.L.M.: Conceptualization, software, investigation, methodology, supervision, writing-original draft, and editing.

## Competing interests
The authors declare no competing interests.

## Additional information

**Peer review information** : *Nature Communications* thanks Vijay Sankaran and the other anonymous reviewer(s) for their contribution to the peer review of this work. A peer review file is available.

