## [Transparent Peer Review file · Nature Communications]

Developmental hematopoietic stem cell variation explain clonal hematopoiesis later in life

Corresponding Author: Dr Adam MacLean

Version 0:

Reviewer comments:

Reviewer #1

(Remarks to the Author)

In this paper, the authors point to previous findings that clonal hematopoiesis becomes increasingly common with aging, often in the absence of detectable driver mutations, and that serial blood sampling and modeling of fitness effects has previously implied a model where weak fitness advantages accrue over long time periods, enabling clonal expansion. The authors argue that this weak selection model is problematic because many cells with a weak advantage would be randomly lost. They propose that these fitness-increasing variants may occur during embryonic development, which enables them to expand during this initial phase so they are at less risk of dropping out of the population. They then test this hypothesis by modeling the correlations between pairs of individuals across "fluctuating" methylation sites at different ages, applying a previously published method that uses selected CpG sites which have highly variable methylation status across the population and that are unlikely to be actively regulated, as markers of lineage. They compare the simulations to real pairs, cleverly using public methylation data from both monozygotic and dizygotic twins and random individuals, reasoning that monozygotic twins who also share a placenta will share more clonal variants, enabling them to test the model in different settings.

Overall, the authors put forward a nice hypothesis and the modeling strategy is overall clear. The modeling decisions also all seem reasonable. The results convincingly show that a simple model of clonal dynamics with weak fitness advantages is compatible with the observed correlations across CpG sites between pairs of individuals. The paper uses existing data in a clever way and puts forward clear, simple models which are useful for the clonal hematopoiesis field to move forward, and which are justifiable by previous data and their modeling results. However, there are some assumptions and questions about the model that would be nice for the authors to address:

- The authors state that "weak selection is problematic to explain CH because most somatic cells that acquire such a weak selective advantage would be randomly lost and therefore never attain detectable frequencies." Why would most HSCs with subtle fitness advantages be replaced? My understanding is that the hematopoietic stem cell population is relatively stable across the lifespan, such that mutations in HSCs would be likely to persist, especially those which confer a subtle fitness advantage that aid these cells in self-renewal. Please provide more support for this assertion, which seems to be fundamental to the hypothesis that embryonic variation and fitness advantage is needed.

- An important assumption made here is constant fitness throughout life. A clear alternate model is that there is no/less selection on these clones in the early environment, and that selection coefficients change/grow in magnitude during aging. Could the authors explore this possibility? If this would make the modeling much more complicated, it may be something to explore in future work, but should at least be discussed further in the text.

- Related to the above point, recent studies on clonal expansions with aging have suggested the presence of multiple independent clonal expansions with aging (dois: 10.1038/s41586-022-04786-y, 10.1038/s41586-024-07066-z). To what extent can clonal selection be altered through potential clonal competition, as well as attrition of other HSC clones. While this may be challenging to model, given these recent observations, this should be discussed further.

- Is this positive or negative frequency-dependent growth? Please clarify the frequency-dependent growth parameter more in the main text.

- What are some possible explanations for the early-acquisition clonal fitness advantages early in life, given that they do not have somatic changes?
- Does random initialization of the CpG sites fit what we know about the wave of methylation across the genome and the initial correlations seen across early-life pairs?
- Would monozygotic twins not be expected to share more similar bone marrow microenvironments and other factors affecting clonal changes during the lifespan? How would this affect the modeling?
- In the discussion, the authors mention “genetic heterogeneity of HSCs.” It would also be worth discussing germline genetic variation that appears to alter risk for both developing clonal hematopoiesis and myeloid malignancies that might impact some of these properties (e.g., [dois: 10.1038/s41586-022-05448-9](https://doi.org/10.1038/s41586-022-05448-9), [10.1038/s41586-020-2786-7](https://doi.org/10.1038/s41586-020-2786-7), [10.1038/s41467-023-41315-5](https://doi.org/10.1038/s41467-023-41315-5)).

(Remarks on code availability)
Appropriately annotated code.

Reviewer #2

(Remarks to the Author)

In the manuscript „Developmental hematopoietic stem cell variation explains clonal hematopoiesis later in life” Kreger and colleagues adapt the concept of fluctuating CpG site methylation to trace clonality in the hematopoietic system using twin study datasets and to assess the impact of different selection biases on clonal outgrowth. The manuscript tackles an important question as it is still not clear how clonal hematopoiesis is established in the context of mutations conferring weak selective advantage. The present study addresses this question by analyzing previously published twin datasets. The idea of lineage tracing using fluctuating CpG sites in these datasets adds novelty to previous studies. Based on their findings the authors propose clonal HSC dominance in aged individuals arises from clones that are established before birth and which have a weak selective advantage over other clones.

I am not a specialist in mathematical modelling and therefore cannot judge on the mathematical details of the authors’ model, but based on my knowledge in the field of hematopoiesis and epigenetics, I can say that the authors’ findings provide significant but provocative novel insights into the biology of the aging process of hematopoiesis. However, the present manuscript falls short in considering and ruling out alternative explanations for their observations, which is why I feel that the authors should provide additional analyses and experiments to prove that their interpretation of the data is likely to be correct.

Here are my detailed comments:

Major:

- In their manuscript, the authors talk a lot about “clonal hematopoiesis”. However, “clonal hematopoiesis” is typically defined as “the presence of specific, cancer-associated somatic mutations in hematopoietic cells in the absence of a hematological malignancy or other clonal disorder” (Ahamad et al, Annual Review of Medicine 2023). The analyses performed in the present work rather relate to physiological changes in clonal diversity of the hematopoietic compartment and this should not be confused with “clonal hematopoiesis”.
- It would be interesting though to see how individuals with confirmed clonal hematopoiesis (e.g. based on mutational patterns) behave with respect fCpG methylation patterns and if the clonal fractions estimated by fCpG methylation matches the size of genetic clones.
- Individuals with genetically defined clonal hematopoiesis frequently present mutations affecting the epigenetic machinery. Can the concept of clonal tracing using fCpGs confidently used in this setting? This would be important to test in order to judge whether the developed model could still be used to infer clonal dynamics in these individuals?
- The authors did not rule out that the fCpGs they investigate are associated with SNPs. Strength of correlation fCpG methylation could simply be a different way of discriminating genotypes rather than reflecting HSC clones.
- The methylation data sets use blood samples (and not purified HSCs) to assess HSC clone abundance. The authors did not establish in how far the fCpGs they use are affected by cellular composition of the samples analyzed. The mature cell types present in blood are progeny from the HSCs present in each of the individuals investigated, however, the number of cell divisions needed for a cell to differentiate from HSC to any given blood cell type is not precisely known and likely differs across cell types. In addition, mature blood cell types show huge differences regarding their life span, and hence, their turnover. I suspect that this should have differential effects on the stability of methylation patterns “inherited” from the parental HSC. The authors don’t seem to have considered this option.
- The model assumptions made are not reflecting physiological properties of the human hematopoietic system. E.g. the size of the HSC pool is does not correspond to what is actually likely the case in humans. In addition, the assumption that the

HSC pool size is constant over the human life span is not correct. How do changes in the size of the HSC pool (initially as well as dynamic changes during life time) affect the model?

- Lines 74-80: Can the authors verify their statement that higher fCpG correlation likely represents twins with shared circulation? This is currently purely speculative and should not be interpreted like this without supporting data.

- Lines 82-84: How can the authors explain that DZ twins have a stronger decline in clonal relatedness than unrelated individuals?

- How would the model predictions change if more than 10 clones are assumed? And how many clones could realistically be detected using array methylation data given that DNA methylation changes below 5% are likely due to technical noise.

Minor:

- Lines 33-35: Please add a reference

- Figure 1A: From the figure legend it is not entirely clear to me what is plotted.

- Figure 1B: Please add a figure legend for the color scheme. The color scheme of the bar plot cannot be interpreted from either the figure itself or the legend.

(Remarks on code availability)

Reviewer #3

(Remarks to the Author)

(Remarks on code availability)

Reviewer #4

(Remarks to the Author)

While the topic of the paper seems interesting, I cannot assess the novelty and robustness of the paper in its current form. I identified several issues with the figures presented.

Specifically, in Figure 4, panels D-G appear to be identical copies of the same plots, despite the legend indicating they are from different simulations. Additionally, I have concerns about panels A-C of the same figure, as the middle panels seem to contain identical points, except for the black line, which is not explained in the figure description. A similar issue is observed in Figure 5, panels A-C, middle panel. Due to these concerns, I am unable to provide a thorough review of the paper's quality at this time.

(Remarks on code availability)

Version 1:

Reviewer comments:

Reviewer #1

(Remarks to the Author)

I have carefully reviewed the revised manuscript. The authors have done an outstanding job of addressing all of my concerns.

(Remarks on code availability)

Clearly presented.

Reviewer #2

(Remarks to the Author)

I thank the authors for providing a detailed response to my and my colleagues' comments. The added explanations significantly improve the manuscript and provide better insights into the authors' understanding of the data. However, I have

two remaining questions that have yet to be addressed:

1) In order to confirm that their model provides meaningful and relevant results the authors should perform their clone size estimates in data from patients with genetically defined CHIP. There is published DNA methylation data from individuals with genetically defined CHIP which should be accessible to the authors of the present manuscript (e.g. PMID36097025). Linking epigenetically defined clones with genetically defined clones would be important to validate the model outputs and to correlate their findings to actual clonal HSC fractions found in affected individuals.

2) The authors clarify their approach to SNP filtering in the methods section and refer to the description in S1. However, as far as I can see, in S1 the authors' SNP filtering is described as being exclusively based on DNA methylation values present in their dataset. For a sanity check, I would in addition ask for a formal demonstration of a lack of overlap of selected CpG sites with any known common SNPs (e.g. based on dbSNP).

(Remarks on code availability)

Sufficiently documented Jupyter notebooks!

Reviewer #3

(Remarks to the Author)

(Remarks on code availability)

Reviewer #4

(Remarks to the Author)

The authors have addressed my comment.

(Remarks on code availability)

Version 2:

Reviewer comments:

Reviewer #2

(Remarks to the Author)

The authors replied in detail to my open questions and have provided detailed additional analyses. I have no further comments or criticism and would be happy to see this interesting study published.

(Remarks on code availability)

./.

Reviewer #3

(Remarks to the Author)

(Remarks on code availability)

Reviewer 1 (Remarks to the Author):

In this paper, the authors point to previous findings that clonal hematopoiesis becomes increasingly common with aging, often in the absence of detectable driver mutations, and that serial blood sampling and modeling of fitness effects has previously implied a model where weak fitness advantages accrue over long time periods, enabling clonal expansion. The authors argue that this weak selection model is problematic because many cells with a weak advantage would be randomly lost. They propose that these fitness-increasing variants may occur during embryonic development, which enables them to expand during this initial phase so they are at less risk of dropping out of the population. They then test this hypothesis by modeling the correlations between pairs of individuals across “fluctuating” methylation sites at different ages, applying a previously published method that uses selected CpG sites which have highly variable methylation status across the population and that are unlikely to be actively regulated, as markers of lineage. They compare the simulations to real pairs, cleverly using public methylation data from both monozygotic and dizygotic twins and random individuals, reasoning that monozygotic twins who also share a placenta will share more clonal variants, enabling them to test the model in different settings.

Overall, the authors put forward a nice hypothesis and the modeling strategy is overall clear. The modeling decisions also all seem reasonable. The results convincingly show that a simple model of clonal dynamics with weak fitness advantages is compatible with the observed correlations across CpG sites between pairs of individuals. The paper uses existing data in a clever way and puts forward clear, simple models which are useful for the clonal hematopoiesis field to move forward, and which are justifiable by previous data and their modeling results.

Response: We appreciate the reviewer’s enthusiasm and thoughtful comments on our work.

However, there are some assumptions and questions about the model that would be nice for the authors to address:

- The authors state that “weak selection is problematic to explain CH because most somatic cells that acquire such a weak selective advantage would be randomly lost and therefore never attain detectable frequencies.” Why would most HSCs with subtle fitness advantages be replaced? My understanding is that the hematopoietic stem cell population is relatively stable across the lifespan, such that mutations in HSCs would be likely to persist, especially those which confer a subtle fitness advantage that aid these cells in self-renewal. Please provide more support for this assertion, which seems to be fundamental to the hypothesis that embryonic variation and fitness advantage is needed.

Response: Yes indeed the hematopoietic stem cell population is relatively stable across lifespan, however a classic result from evolutionary biology tells us that the fixation probability of a beneficial mutation (with fitness advantage s) introduced into a large, relatively constant population is $2s$ (Haldane, 1927, A Mathematical Theory of Natural and Artificial Selection). This

result has been further supported in other mathematical studies (Moran, 1962, The Statistical Processes of Evolutionary Theory; Kimura, 1962, PMID:14456043; Patwa, 2008, PMID:18664425; Kreger, 2023, PMID:36514852). Thus, for HSCs in the bone marrow niche, when individual cells acquire a weak selective advantage ($s > 0$ but close to 0), there is a high probability that it will be randomly lost and not reach large population sizes within a reasonable time span. However, such a weakly advantageous mutant generated during a time of exponential growth experiences very different dynamics. At such a time all clones are expanding (e.g. during development) and so there is a much higher probability that mutants can attain detectable frequencies throughout life.

Additional clarification of this point has been added to the Introduction section of the manuscript. In addition to the Introduction, we also note in the original submission in lines 199-207: *“as long as the fittest clones were not lost due to random cell loss... even if the fittest clone starts with few initial cells at birth relative others, over many decades it will grow to dominate, even in the large size of the HSC pool (the classical probability of fixation in a large population for a single cell with fitness advantage s is given by $2s$ (Haldane, 1927, A Mathematical Theory of Natural and Artificial Selection). The timing of when variants arise under strong selection does not greatly affect HSC clonal dynamics in later life since a loss of clonal diversity is effectively ensured. Under this model, the timing of when variants arise would affect only the time at which the clone takes over (earlier generation/faster initial growth resulting in earlier fixation).”*

- An important assumption made here is constant fitness throughout life. A clear alternate model is that there is no/less selection on these clones in the early environment, and that selection coefficients change/grow in magnitude during aging. Could the authors explore this possibility? If this would make the modeling much more complicated, it may be something to explore in future work, but should at least be discussed further in the text.

- Related to the above point, recent studies on clonal expansions with aging have suggested the presence of multiple independent clonal expansions with aging (doi: 10.1038/s41586-022-04786-y, 10.1038/s41586-024-07066-z). To what extent can clonal selection be altered through potential clonal competition, as well as attrition of other HSC clones. While this may be challenging to model, given these recent observations, this should be discussed further.

Response: This is an interesting suggestion. As noted, in the manuscript we assumed that selection throughout life (i.e. from birth) is constant. An alternative assumption as the reviewer notes could be that selection coefficients increase in magnitude as we age. We have explored this possibility in the revision. In summary: selection coefficients that increase as we age are less consistent with the HSC data in our manuscript. The varying-selection-coefficients model leads more often to scenarios in which we see (harmful) driver mutations that are likely causing large clonal expansions that will take over the niche, i.e. result in malignancy.

In a new figure shown below (new Fig. S10 in the revision) we simulate pairs of monozygotic twins under the assumptions that:

1. clones are neutral during development ($s_i = 0$ for all i clones), and

2. the fitness of each clone throughout life is (positively) proportional to its frequency. Here we have the fitness of each clone is given by $1 + s_i(t) = 1 + F_i(t)/10$ for $i = 1, 2, \dots, N_{clones}$ where $F_i(t)$ represents the frequency of each clone at time t , which is updated at the beginning of each discrete time step. For more on positive frequency-dependent selection see for example Svensson, 2018, PMID:31417612, Lande, 1976, PMID:28563044.

In this way, clones that are more prevalent have growing fitness advantages with a maximum of a 10% advantage in the case a single clone fixates in the population. In panel A we show an example of a simulation where the same clone takes over in both twins. In panel B we show an example of a simulation where different clones take over in the two twins. Individual clones tend to take over due to their growing fitness advantages which drives the Pearson correlation coefficient to one (same clone takes over) or zero (different clones take over). This is similar to the constant strong selection scenario (Fig. 4C,F in the main text), and does not match what we see in the data (Fig. 1F in the main text). This is discussed further in new section S7 “Time-dependent selection during aging reduces clonal diversity” in the Supplementary Information as well as in the Results section of the main text.

As more detailed information about clonal hematopoiesis and clonal compositions becomes available (e.g. Mitchell, 2022, PMID:35650442; Weng, 2024, PMID:38253266; Watson, Evolutionary dynamics in the decades preceding acute myeloid leukaemia, 2024, bioRxiv) we can build more detailed models that potentially can include time-dependent fitness effects, negative frequency-dependent selection (see e.g. Christie, 2023, PMID:37484931), effects of microenvironments, etc. A discussion of these future topics has been added to the Discussion section of the manuscript.

Also related to the reviewer’s second comment, we note that the current model does permit (and we do observe) the presence of multiple independent clonal expansions with aging. Our argument is that these clones arise during development and then reach notable frequencies many decades later. The model can explicitly show clonal competition and attrition of some HSC clones (e.g. see left panels of figure below).

Figure S10: **Growing fitness coefficients during aging reduces clonal diversity.** Simulations shown for monozygotic twins ($N_{split} = 36$). We assume clones are neutral during development and the fitness of each clone during life is one plus the frequency of the clone divided by 10. A: The same clone takes over in both twins and the Pearson correlation coefficient trends toward one. B: Different clones take over in the twins and the Pearson correlation coefficient trends toward zero.

- Is this positive or negative frequency-dependent growth? Please clarify the frequency-dependent growth parameter more in the main text.

Response: We assume positive frequency-dependent growth during development. That is, clones with higher frequencies are more likely to be chosen for reproduction in subsequent iterations. This has been clarified in two places in the main text: the “Stem cell variation arising during development explains the blood clonal composition at birth” paragraph in the Results section, as well as in the “Variation in development” paragraph of the Methods section. As noted in the previous response, for more on positive frequency-dependent selection see for example Svensson, 2018, PMID:31417612, Lande, 1976, PMID:28563044.

- What are some possible explanations for the early-acquisition clonal fitness advantages early in life, given that they do not have somatic changes?

Response: In our manuscript, we show that it is likely that clonal dynamics are driven by weak selection that becomes evident with aging. Therefore, fitness differences between clones are small and are not driven by known driver mutations. Zink et al. (2017, PMID:28483762) provides evidence for the lack of known driver mutations in most cases of clonal hematopoiesis and speculated that epigenetic changes are potentially responsible for small fitness differences between clones. Our study allows us to examine clonal frequencies before birth and throughout

life in twins/unrelated individuals, but does not identify the underlying selection mechanisms, which is still an open question in clonal hematopoiesis.

Additional discussion of this point has been added to the Introduction section of the manuscript.

- Does random initialization of the CpG sites fit what we know about the wave of methylation across the genome and the initial correlations seen across early-life pairs?

Response: We assume that each fCpG site for each clone is initially randomly either homozygously unmethylated or homozygously methylated. This is because during the early stages of embryogenesis, the inherited methylation patterns from parental gametes are largely erased before a large wave of de novo methylation remodels the entire genome, resulting in a bimodal methylation distribution (Cedar, 2012, PMID:22404632, Gabbutt, 2022, PMID:34980912). This is noted in the model description in the Methods section of the manuscript.

- Would monozygotic twins not be expected to share more similar bone marrow microenvironments and other factors affecting clonal changes during the lifespan? How would this affect the modeling?

Response: While monozygotic twins likely share more similar bone marrow microenvironments and other factors post birth, we assume that these factors are negligible compared to shared circulation of blood and HSCs during development. We think this is a reasonable assumption given the large difference in similarity between shared blood in development (highly similar) vs shared microenvironmental factors throughout life, the latter involving many intermediary steps from transcription and translation to secretion of signaling molecules, diffusion, spatial organization of the tissue, etc. Thus in the current work we did not model these factors. It would be interesting to consider the effects of shared HSC microenvironmental and other factors in modeling efforts in future work. Discussion to this extent has been added to the manuscript.

- In the discussion, the authors mention “genetic heterogeneity of HSCs.” It would also be worth discussing germline genetic variation that appears to alter risk for both developing clonal hematopoiesis and myeloid malignancies that might impact some of these properties (e.g., doi: 10.1038/s41586-022-05448-9, 10.1038/s41586-020-2786-7, 10.1038/s41467-023-41315-5).

Response: In the current model, we use fCpG sites (as opposed to somatic mutations) as a way to measure clonal diversity throughout development and life. As noted by the reviewer, there is germline genetic variation that has been shown to affect clonal hematopoiesis. Especially due to our focus on modeling development starting from a small number of HSCs, future models building on this current work ought to consider germline genetic variation and

their influence on healthy hematopoiesis in development. Discussion of germline variants as risk factors for clonal hematopoiesis and myeloid malignancies has been added to the Discussion section of the manuscript.

Reviewer 1 (Remarks on code availability):

Appropriately annotated code.

Reviewer 2 (Remarks to the Author):

In the manuscript „Developmental hematopoietic stem cell variation explains clonal hematopoiesis later in life” Kreger and colleagues adapt the concept of fluctuating CpG site methylation to trace clonality in the hematopoietic system using twin study datasets and to assess the impact of different selection biases on clonal outgrowth. The manuscript tackles an important question as it is still not clear how clonal hematopoiesis is established in the context of mutations conferring weak selective advantage. The present study addresses this question by analyzing previously published twin datasets. The idea of lineage tracing using fluctuating CpG sites in these datasets adds novelty to previous studies. Based on their findings the authors propose clonal HSC dominance in aged individuals arises from clones that are established before birth and which have a weak selective advantage over other clones.

I am not a specialist in mathematical modelling and therefore cannot judge on the mathematical details of the authors’ model, but based on my knowledge in the field of hematopoiesis and epigenetics, I can say that the authors’ findings provide significant but provocative novel insights into the biology of the aging process of hematopoiesis. However, the present manuscript falls short in considering and ruling out alternative explanations for their observations, which is why I feel that the authors should provide additional analyses and experiments to prove that their interpretation of the data is likely to be correct.

Here are my detailed comments:

Major:

- In their manuscript, the authors talk a lot about “clonal hematopoiesis”. However, “clonal hematopoiesis” is typically defined as “the presence of specific, cancer-associated somatic mutations in hematopoietic cells in the absence of a hematological malignancy or other clonal disorder” (Ahmad et al, Annual Review of Medicine 2023). The analyses performed in the present work rather relate to physiological changes in clonal diversity of the hematopoietic compartment and this should not be confused with “clonal hematopoiesis”.

Response: We thank the reviewer for their thoughtful comments and positive feedback on the appeal and novelty of our work. Regarding the definition of “clonal hematopoiesis,” we appreciate the need to clarify our use of the term in the manuscript. We do believe that a broader definition of clonal hematopoiesis (CH) as “clonal expansions of HSCs” is appropriate, and would note that we make a distinction between CH and CHIP (clonal hematopoiesis of indeterminate potential) as is defined by the quote above from (Ahmad, 2023, PMID:36450282). Whereas CHIP is clinically focused, we believe that CH is an appropriate term for the observed changes in clonal diversity in the HSC niche.

We would also note that, while CH often involves specific, cancer-associated somatic mutations, there are many other ways to detect CH other than via mutation, including karyotype changes

(loss of Y chromosome), skewed X-chromosome inactivation, or via flow cytometry the loss of antigens such as in paroxysmal nocturnal hemoglobinuria (PNH).

We have added clarification of this to the first paragraph of Introduction section of the main text, specifically noting that:

“Whereas in some cases driver mutations can be found, often CH is lacking identifiable drivers that can explain their expansions. CH is often identified by specific somatic mutations. Here we take a broader view, defining CH as clonal expansions in the HSC pool that lead to a loss of stem cell diversity. Consistent with the frequent lack of strong driver mutations, serial observations of CH are often compatible with weak selection (potentially driven by epigenetic changes) because clone sizes are stable or expanding slowly over many years.”

- It would be interesting though to see how individuals with confirmed clonal hematopoiesis (e.g. based on mutational patterns) behave with respect fCpG methylation patterns and if the clonal fractions estimated by fCpG methylation matches the size of genetic clones.

Response: This is a very important point, and the subject of active ongoing research. While in the current work we focused on mutation-agnostic dynamics of HSCs, investigating the methylation patterns and genetics of HSCs is an important future step. Indeed, other groups in addition to ours are also studying these questions. Building on the original work of (Gabbutt et al., 2022, PMID:34980912), see e.g. preprints and poster abstracts:

- Mallo et al, 2024, Abstract PR009: Tick tock trees: Reconstructing the evolutionary dynamics of human tissues using fluctuating methylation clocks, https://aacrjournals.org/cancerres/article/84/3_Supplement_2/PR009/733443 - method to produce estimates like ancestral relationships between sampled clonal populations, their calendar divergence times, the number of effective evolutionary niches, and how some of these parameters change over time using fluctuating methylation
- Gabbutt et al, 2023, Evolutionary dynamics of 1,976 lymphoid malignancies predict clinical outcome, <https://www.medrxiv.org/content/10.1101/2023.11.10.23298336v2> - method called EVOFLUX, based upon natural DNA methylation barcodes fluctuating over time, that quantitatively infers evolutionary dynamics using only a bulk tumor methylation profile as input
- Schenck et al, 2022, Abstract 634: Mutation agnostic diagnosis of clonal hematopoiesis of indeterminate potential using fluctuating methylation clocks, https://aacrjournals.org/cancerres/article/82/12_Supplement/634/699519 - machine learning method that allows us to diagnose CHIP without DNA sequencing

We have added the following to the Discussion section of the manuscript:

“It would also be interesting to analyze how individuals with confirmed clonal hematopoiesis behave with respect to fCpG methylation patterns and if the clonal fractions estimated by methylation dynamics corresponds with the sizes of genetic clones.”

- Individuals with genetically defined clonal hematopoiesis frequently present mutations affecting the epigenetic machinery. Can the concept of clonal tracing using fCpGs confidently used in this setting? This would be important to test in order to judge whether the developed model could still be used to infer clonal dynamics in these individuals?

Response: The reviewer is right to point out that the most common genes mutated in CH (DNMT3A, TET2, ASXL1) are associated with DNA methylation and chromatin remodeling. Thus it is important to consider whether mutations in these genes would change the fCpG dynamics. For example, a mutation in DNMT3A could lead to hypomethylation. Through extensive studies in the original fCpG paper (Gabbutt, 2022 PMID:34980912), clonal tracing via fCpGs was found to successfully recapitulate the dynamics of stem cells in intestinal crypts, endometrial glands, and in the blood. In particular, these common mutations did not significantly alter the methylation dynamics at the fCpGs because methylation remains balanced and around 50% in blood with aging, CHIP, and in acute myeloid leukemia (Gabbutt, 2022 PMID:34980912). Methylation in our samples is also balanced around 50%. Thus, while we have not fully elucidated the interplay between fCpG dynamics and CH-associated somatic mutations, empirically the fCpG method for clonal tracing has been successfully applied across various tissues.

Future work could involve sequencing the blood of monozygotic twins pairs throughout life and identifying if methylation patterns are still concordant when one twin has a somatic mutation (e.g. DNMT3A) and the other twin does not. This would identify whether or not identical somatic mutations are behind the concordance we observe in methylation dynamics with monozygotic twins. However, as described in (Hansen, 2020, PMID:31697811), sequencing of blood in older monozygotic twins does not show identical somatic mutations (which is consistent with most somatic mutations accumulating during aging after birth). Therefore, it is extremely unlikely that identical somatic mutations are behind the concordance in methylation patterns that we observe in monozygotic twins.

- The authors did not rule out that the fCpGs they investigate are associated with SNPs. Strength of correlation fCpG methylation could simply be a different way of discriminating genotypes rather than reflecting HSC clones.

Response: We can clarify that our selection of “neutral CpG” sites did indeed check for this and we remove any sites that showed signs of association with known SNPs. The detailed methodological steps for how this is performed are provided in the Supplementary Information (section 1). We have also added clarification of this point to the Methods section in the main text.

- The methylation data sets use blood samples (and not purified HSCs) to assess HSC clone abundance. The authors did not establish in how far the fCpGs they use are affected by cellular composition of the samples analyzed. The mature cell types present in blood are progeny from

the HSCs present in each of the individuals investigated, however, the number of cell divisions needed for a cell to differentiate from HSC to any given blood cell type is not precisely known and likely differs across cell types. In addition, mature blood cell types show huge differences regarding their life span, and hence, their turnover. I suspect that this should have differential effects on the stability of methylation patterns “inherited” from the parental HSC. The authors don’t seem to have considered this option.

Response: We use an “HSC-specific” fCpG list, see point 5 in Supplementary Information section 1. We have added clarification of this point to the Methods section in the main text.

- The model assumptions made are not reflecting physiological properties of the human hematopoietic system. E.g. the size of the HSC pool is does not correspond to what is actually likely the case in humans. In addition, the assumption that the HSC pool size is constant over the human life span is not correct. How do changes in the size of the HSC pool (initially as well as dynamic changes during life time) affect the model?

Response: In the model we do indeed make simplifying assumptions that do not always correspond directly to human hematopoiesis. The key assumptions as the reviewer describes are that: 1) the size of the HSC pool throughout life is smaller than in reality (see lines 284-286 in the Discussion section), and 2) a constant HSC population size is used during life (see lines 101-102 in the Results section). Fig. 2B in the manuscript provides a visualization of these assumptions. When modeling complex biological systems it is essential to identify core components of the system that are explicitly modeled and then make assumptions where one can simplify out other details (Ingalls, *Mathematical Modelling in Systems Biology*, 2012; Nowak, *Evolutionary Dynamics*, 2006). Specifically regarding the assumptions surrounding the HSC population size:

1) the number of HSCs in the model (5,000) is smaller than in reality (~25,000 - $1e6$, PMID:34742656).

We developed a mathematical model of methylation dynamics at the resolution of individual fCpG sites in single cells, with a time step of one day over a time range of 100 years. Simulating the stochastic dynamics of this model is computationally quite costly. Thus, we chose a HSC population size that allowed us to perform model simulations with relative ease. Increasing the population size from 5,000 by a factor of around 10x would have a negligible effect on the initial clone distribution (see e.g. Fig. 3A-B, left panels: clonal frequencies do not change after pop. size of approx. 1,000 cells). After birth, increasing the population size will result in a “slowing down” of the dynamics, but will not fundamentally alter clonal competition. I.e. it will take a longer time to eliminate a non-advantageous clone of low frequency in a large population. For example, consider clones of the same frequency, 1,000 out of 100,000 cells (large population) vs. 1 out of 100 cells (small population): the former is likely to persist for longer (Moran, *The statistical processes of evolutionary theory*, 1962).

2) The HSC population size is constant throughout life.

Variation in the HSC population size during life does occur, most notably the HSC pool expands during aging. However, even if the HSC pool increased in size by 50% (1.5x), this change would be small relative to changes during the growth phase of development (100x). Therefore, changes in the HSC population size during life will likely have negligible effects on clonal frequencies and thus Pearson correlation coefficients between individuals.

Discussion of these points has been added to the manuscript, along with careful exposition of other simplifying assumptions made in the model in the Discussion.

- Lines 74-80: Can the authors verify their statement that higher fCpG correlation likely represents twins with shared circulation? This is currently purely speculative and should not be interpreted like this without supporting data.

Response: In monozygotic twins, there is quite extensive evidence that about 2/3 of twins share prenatal blood circulation (monochorionic placenta) and about 1/3 do not (dichorionic placenta) (Marceau, 2016, PMID:26944881). As previously noted in the manuscript (lines 74-80), there is greater heterogeneity in MZ twin pairs than DZ or unrelated, with approximately 2/3 of MZ twin pairs showing higher HSC correlations at birth than the remaining 1/3 of MZ twin pairs (Fig. 1 in the main text). This bimodal MZ twin pair correlation at birth suggests that the pairs with higher correlations represent twins with shared prenatal circulation. However, this observation is currently speculative as we cannot discern characteristics of shared vs. non-shared circulation from the datasets we have. We have clarified this on line 80: that future experiments are needed to confirm that monozygotic twins that share prenatal blood circulation have a higher Pearson correlation coefficient at birth.

- Lines 82-84: How can the authors explain that DZ twins have a stronger decline in clonal relatedness than unrelated individuals?

Response: One explanation for this could be that dizygotic twins have a higher Pearson correlation coefficient at birth, but that later in life they lose the “additional twin” correlation effect observed in the data and thus appear to be most similar to unrelated individuals. This would result in a more negative slope in the line of best fit for dizygotic twins compared to unrelated individuals, as we see (Fig. 1F in the main text). This concordance being lost with aging in dizygotic but not monozygotic twins further supports the idea that variation acquired before birth can lead to clonal hematopoiesis later in life.

Monozygotic twins and dizygotic twin studies are interesting because traits usually depend on both environmental and inherited factors. Generally monozygotic and dizygotic twins share more similar environments than unrelated individuals, so twins in general are more concordant for many parameters than unrelated pairs (Hagenbeek, 2023, PMID:37188734). We speculate that the more similar environment at birth for all twins leads to a higher concordance at birth. This could explain why monozygotic twins have the highest initial Pearson correlation

coefficient, followed by dizygotic twins, followed by unrelated individuals. Since this concordance is lost during life for dizygotic twins, there is a greater decline (more negative slope) in life.

- How would the model predictions change if more than 10 clones are assumed? And how many clones could realistically be detected using array methylation data given that DNA methylation changes below 5% are likely due to technical noise.

Response: We have performed simulations with larger numbers of clones, see Fig. 2C and the new Fig. S4 in Supplementary Information section 4. In Fig. 2C in the main text, we show how calculation of the N_{split} parameter is made, e.g. if we assume $N_{clones} = 20$ then N_{split} is approximately for 20 for unrelated individuals, 30 for dizygotic twins, and 75 for monozygotic twins. Furthermore, Fig. S3 shows the effect of the number of clones (N_{clones}): when there is a larger number of clones, the clonal distributions of the individuals under comparison will be less similar as there is more variability in the clonal distributions during development. This results in a lower Pearson correlation coefficient as it becomes increasingly likely that different clones will rise to prominence between individuals (see Supplementary Information section 4).

New Fig. S4 (included below) shows simulations with $N_{clones} = 20$ for monozygotic twins, dizygotic twins, and unrelated individuals with weak selection, similarly to main text Fig. 5. Overall, there are no major differences between using 10 (Fig. 5) or 20 (Fig. S4) clones and there will not be major differences as long as N_{clones} is still significantly less than the total number of cells N_{cells} .

In summary, increasing the number of cell clones results in two effects: 1) different values of N_{split} need to be calculated (N_{clones} and N_{split} are positively correlated) and 2) overall the Pearson correlation coefficients may be slightly lower as there is an increased likelihood that different clones dominate the dynamics.

Figure S4: **Simulations for number of cell clones** $N_{clones} = 20$. Simulations shown for $N_{clones} = 20$ with weak selection ($a = 0.05$ and $\theta = 0.01$). **A-C**: Clone growth frequency plots for both individuals during development and life (dashed vertical line represents N_{split}), Pearson correlation coefficient, and β distributions at 0, 50, and 100 years of life. A: MZ twins, $N_{split} = 75$. B: DZ twins, $N_{split} = 30$. C: Unrelated individuals, $N_{split} = 20$. **D-F**: Results from 50 simulations are shown (dashed lines are individual simulations and solid lines are mean trajectories). D: MZ twins. E: DZ twins. F: Unrelated individuals.

For the second question, we do not know the maximum number of cell clones that can be detected using array methylation data. This could be an interesting experimental study for future work. Using a mathematical framework, we can simulate any number of cell clones.

Minor:

- Lines 33-35: Please add a reference

Response: We have added the reference: *Hansen, J. W. et al. Clonal hematopoiesis in elderly twins: concordance, discordance, and mortality. Blood 135, 261–268 (2020). PMID: 31697811*

To support the claim that CH driver mutations are usually not concordant between MZ twins, indicating that driver mutations arise after birth and that MZ twins do not share a predisposition for specific mutations.

- Figure 1A: From the figure legend it is not entirely clear to me what is plotted.

Fig. 1A is a cartoon of clonal diversity in the HSC compartment between twins from development through life. We have updated the figure caption to clarify this.

- Figure 1B: Please add a figure legend for the color scheme. The color scheme of the bar plot cannot be interpreted from either the figure itself or the legend.

Response: We thank the reviewer for these suggestions to improve the clarity of Fig. 1. We have added legends to Figures 1A and 1B as well as added more detail to the figure caption.

Reviewer 3 (Remarks to the Author):

Reviewer 4 (Remarks to the Author):

While the topic of the paper seems interesting, I cannot assess the novelty and robustness of the paper in its current form. I identified several issues with the figures presented. Specifically, in Figure 4, panels D-G appear to be identical copies of the same plots, despite the legend indicating they are from different simulations. Additionally, I have concerns about panels A-C of the same figure, as the middle panels seem to contain identical points, except for the black line, which is not explained in the figure description. A similar issue is observed in Figure 5, panels A-C, middle panel. Due to these concerns, I am unable to provide a thorough review of the paper's quality at this time.

Response: We appreciate the reviewer's time in reading our manuscript. With regards to the concern raised about identical figure panels: we would like to clarify that in each panel the *data points* plotted are the same and the *simulation points* plotted are different.

Throughout this study, we used a single dataset (that was compiled from many publicly available sources; see Table 1 in the manuscript). We used this compiled dataset to compare against various computational model scenarios represented by different model simulations.

In each of the figure panels in Figs. 4 and 5 (and SI figures), representations of the entire dataset are plotted (colored points or ribbons) to compare with model simulations (gray or black lines). The model simulations are different in each panel based on the particular scenario considered, as defined in the figure captions. The data in each panel are the same. In the previous version, the figure captions fully defined only the simulations plotted and not the underlying data plotted for comparison. We have updated the figure captions in the revision and have also added clarification of this point in the Results section of the manuscript.

We would also like to note that the raw data underlying each figure panel (both the methylation data and the data points that were generated by each model simulation run) is available on a GitHub repository at: <https://github.com/maclean-lab/scFMC-model> (this page has been online and available since the paper was first submitted). Specifically, the data underlying plots in the manuscript is available in tabular format here: <https://github.com/maclean-lab/scFMC-model/tree/main/data>.

Reviewer #2 (Remarks to the Author):

I thank the authors for providing a detailed response to my and my colleagues' comments. The added explanations significantly improve the manuscript and provide better insights into the authors' understanding of the data. However, I have two remaining questions that have yet to be addressed:

Response: We appreciate the reviewer's time and thoughtful comments on our work.

1) In order to confirm that their model provides meaningful and relevant results the authors should perform their clone size estimates in data from patients with genetically defined CHIP. There is published DNA methylation data from individuals with genetically defined CHIP which should be accessible to the authors of the present manuscript (e.g. PMID36097025). Linking epigenetically defined clones with genetically defined clones would be important to validate the model outputs and to correlate their findings to actual clonal HSC fractions found in affected individuals.

Response: Linking epigenetically-defined HSC clones with genetically-defined clones in individuals with CHIP is a very interesting and important research direction, and is indeed the subject of other current work (e.g. Evolutionary dynamics in the decades preceding acute myeloid leukaemia, bioRxiv, <https://www.biorxiv.org/content/10.1101/2024.07.05.602251v1>, 2024). We note that in this study we use the data from methylation sequencing arrays to calculate average methylation β -distributions and from these we investigate correlations between individuals (twins vs unrelated). In other words, we are studying correlations between average methylation profiles over a set of selected CpG sites. Due to the large number of HSCs and HSC clones in the blood/bone marrow, it is not possible to directly estimate clone sizes from methylation data of HSCs. (As discussed in Gabbutt et al. PMID: 34980912, the bone marrow is predominantly polyclonal, consisting of thousands of stem cells in a relatively well-mixed system, unlike e.g. colonic crypts where drifts to clonality are observed due to the architecture and small number of stem cells present.)

Using our mathematical model, which acts as a bridge between methylation data and the actual clonal dynamics of HSCs, we are able to produce both clone size estimates and methylation β -distributions in individuals by simulating stem cell competition between thousands of single cells from different clones over a 100-year period.

Using sequencing data from individuals with CHIP, we can compare the methylation β -distributions between model and data. Below we show a comparison of the β -distributions between healthy and CHIP individuals. For this comparison, the healthy individuals are sampled from GSE105018 and GSE73115 and the methylation data for individuals with CHIP are from: <https://www.biorxiv.org/content/10.1101/2022.08.25.505316v2>, 2024), data available on GEO:

GSE210435. (we did not use the data from (PMID: 36097025) as timelines for the approval process to obtain their data range from 4–9 weeks.)

The figure below (see new Fig S11 and new Section S8 in the Supplementary Information) can be compared to Fig. 6C-D of (Gabbutt et al. 2022, PMID: 34980912), where we see that increasing clonality leads to higher variance in the β -distributions. We also can compare both healthy and CHIP methylation distributions with the model we developed in this paper to study HSC dynamics at single-cell level: we see that the CHIP data corresponds to individuals with decreased polyclonality in our model, but not yet with clonal expansions large enough to take over the niche (see Fig. 4 in the current paper for illustrative beta-distributions).

Thus, although we cannot directly compare clone size estimates between model and data, we can compare the fluctuating methylation distributions, and we see that in all cases these are consistent. It is also important to note that we are not aware of any current methylation studies in (monozygotic or dizygotic) twins where one or both twins have genetically defined CHIP. Neither are there to our knowledge any longitudinal studies on twins with CHIP with methylation data. Data from such a study would be ideal to implement our methods/model to directly correlate β -distributions and correlation coefficients in twins with CHIP. However, this is currently not feasible. Future work could also involve testing our model predictions against clonal data/evolutionary predictions from for example (Evolutionary dynamics in the decades preceding acute myeloid leukaemia, bioRxiv, <https://www.biorxiv.org/content/10.1101/2024.07.05.602251v1>, 2024), but this is outside the scope of the current work.

Figure S11: CHIP increases the variance of β -distributions. Histograms show β -distributions from five individuals **A**: with genetically defined CHIP from GSE210435. **B**: non-CHIP 18-year-olds from GSE105018. **C**: non-CHIP 76-year-olds from GSE73115. **D**: non-CHIP 86-year-olds from GSE73115. There is no relation among individuals taken from the same dataset. If more rapid clonal expansions occur (e.g. acute leukemias, major hematopoietic neoplasms), then the variance of the distribution will increase even more and characteristic W-shaped distributions will be observed (Gabbutt 2022, PMID: 34980912, Fig 6).

2) The authors clarify their approach to SNP filtering in the methods section and refer to the description in S1. However, as far as I can see, in S1 the authors' SNP filtering is described as being exclusively based on DNA methylation values present in their dataset. For a sanity check, I would in addition ask for a formal demonstration of a lack of overlap of selected CpG sites with any known common SNPs (e.g. based on dbSNP).

Response: To demonstrate that the concordance between methylation data in identical twins is not driven by SNPs, we compared our list of “neutral” CpG sites with CpG sites that are near known SNPs. To do this, we use Illumina’s publicly available Methylation SNP list (https://support.illumina.com/downloads/infinium_hd_methylation_snp_list.html) which contains 273,660 CpG sites along with their tag SNPs, distance from SNPs, and minor allele frequencies (MAF).

When comparing the 3,918 neutral CpG sites used in our study against the 273,660 in the Illumina list, there are 115 CpG sites that appear in both and have distance 0 from their tag SNP. Of these 115 SNPs:

- 2 have MAF of 50% (and the nearby CpG site is effectively neutral)
- 113 have MAF below 6%,
- 111 have MAF below 5%, and
- 99 have MAF below 1%.

A histogram of the MAF for the SNPs tagging the 115 CpG sites is included below (left panel). The right panel shows the MAF for all SNPs within 51 bp (the maximum distance in the Illumina file) of any CpG site included in our list.

Figure: Minor allele frequency is extremely low for most sites associated with known SNPs. Histograms of MAF of the SNPs tagging neutral CpG sites used in this study. The dashed red line shows a threshold value of 1% and the green line is a threshold of 5%. Left panel: distance 0 bp. Right panel: distance < 52 bp. Most of the tag SNPs exist at a low frequency and it is unlikely that the alternate allele would appear in the data.

A MAF around 5% is a common threshold to differentiate between uncommon/common SNPs (Precision Medicine: Tools and Quantitative Approaches, 2018). We only identified two CpG sites that are tagged by common SNPs (and have MAF 50%).

CpG site	SNP	Minor allele frequency
cg01581781	rs113267937	0.5
cg22454022	rs111595440	0.5

Both of these SNPs have a MAF of 50% and are associated with zero publications according to dbSNP.

The remaining 113 SNPs with low MAF are likely fixed for the reference allele (since the reference allele is nearly fixed in the population) in the data we have sampled from.

It makes sense that if we are selecting for neutral CpGs then any tag SNP would most likely be fixed for the reference allele (uncommon SNP) or very common (MAF of 50%). Because most of the genome is fixed and shared homozygous in all human populations - the SNPs with MAF close to 50% are likely very old, shared, and effectively neutral (or so weakly deleterious that they behave neutrally), and thus can be included in our analysis.

Therefore, we conclude that the presence of identical SNPs is not driving the concordance in blood methylation between identical twins. Common non-neutral SNPs are not associated with any of our neutral CpG sites because our pre-filtering steps (see Methods and Supplementary Information Section 1) get rid of obvious and common SNPs - that are suspicious because their methylation values are distributed around 0, 0.5, and 1.0, instead of range between 0 and 1.

The list of 115 CpG sites that have distance 0 from their tag SNP has been added to the github repository associated with this manuscript here: <https://github.com/maclean-lab/scFMC-model/tree/main/data>. Furthermore, additional explanation that common non-neutral SNPs are not associated with any of our neutral CpG sites has been added to the Supplementary Information Section 1.